# A New Co-Crystal of Synthetic Drug Rosiglitazone with Natural Medicine Berberine: Preparation, Crystal Structures, and Dissolution

**DOI:** 10.3390/molecules25184288

**Published:** 2020-09-18

**Authors:** Xiaoshu Guan, Lan Jiang, Linhong Cai, Li Zhang, Xiangnan Hu

**Affiliations:** 1Department of Medicinal Chemistry, School of Pharmacy, Chongqing Medical University, Chongqing 400016, China; 17082343045@163.com (X.G.); cailinhong123@163.com (L.C.); zhangli3357@163.com (L.Z.); 2College of Environment and Resources, Chongqing Technology and Business University, Chongqing 400016, China; jianglanlucky@163.com

**Keywords:** co-crystal, rosiglitazone, berberine, electrostatic attraction, moisture adsorption, dissolution rate

## Abstract

A co-crystal of rosiglitazone (Rsg) with berberine (Bbr), Rsg-Bbr, was prepared by the solvent evaporation method and characterized. The results showed that the electrostatic attraction existed between the nitrogen anion of rosiglitazone and the quaternary ammonium cation of berberine, and C-H···O hydrogen bonds were formed between Rsg and Bbr. In the crystal structure, rosiglitazone molecules stack into a supramolecular layer through π-π interactions while π-π interactions between berberine cations also result in a similar layer. The co-crystal presented a low moisture adsorption curve in the range of 0−95% relative humidity values at 25 °C. The improved dissolution rate of rosiglitazone in pH = 6.8 buffer solution could be achieved after forming co-crystal.

## 1. Introduction

Although the co-crystal has a history of more than 100 years, it has been used in the field of medicine in the last two decades [1]. The new drug co-crystal is formed by an active pharmaceutical ingredient (API) and another or more cocrystal formers (CCFs) with the action of non-covalent bonds, which could improve many physical and chemical properties, such as solubility and stability [2,3]. It is worth noting that drug-drug co-crystals based on two APIs have attracted new attention in the pharmaceutical field recently [4,5,6]. This multidrug crystal is superior to traditional drug crystal because it is one drug that contains two APIs to achieve the effect of synergistic treatment for specific diseases [7]. Take an example, entresto, the first dual-acting angiotensin receptor-neprilysin inhibitor (ARNi) drug composed of sacubitril and valsartan, has the therapeutic advantages beyond monomer drugs [8,9]. Besides, the formation of reliable supramolecular synthons between two different chemical structures of APIs is essential for the rational design of drug-drug co-crystals [10].

Rosiglitazone (Rsg, Figure 1a), is a highly selective and effective agonist of a peroxisome proliferation-activated receptor-γ (PPARγ), with controlling blood sugar by increasing insulin sensitivity [11,12]. It serves as a first-line thiazolidinedione-containing drug for type 2 diabetes mellitus (T2DM) [13]. Unfortunately, due to the low solubility and high permeability of the molecule itself, Rsg is ranked as a class II molecule by the Biopharmaceutical Classification System (BCS) [14]. The poor solubility of rosiglitazone may affect its effectiveness in the body. Therefore, attempting to tackle the solubility problem and other undesirable properties of Rsg by co-crystallization is worth exploring.

Berberine(Bbr) is one of the most important bioactive isoquinoline alkaloids from the well-known Traditional Chinese Medicine called “Huang Lian”, which has been used in China for over 1000 years [15]. Bbr exists in nature mainly in the form of quaternary ammonium salt, with its chloride salt (Figure 1b) used in the clinical medication. In the research of efficacy, it was found to have various therapeutic effects such as anti-inflammatory [16], inhibiting glucose absorption [17], regulating intestinal flora [18] and protecting intestinal mucosal barrier [19] activities. Interestingly, Bbr has been used to treat type 2 diabetes in recent years [20]. It could increase insulin receptor expression, promote the release and secretion of insulin, and increase the glucose consumption of liver cells [21,22], etc. Due to its feasible hypoglycemic effect, berberine was used in combination with other hypoglycemic drugs to treat type 2 diabetes in clinical drug research [23]. Moreover, its chloride salt has good solubility in water [24], which meets the conditions of co-crystal formers.

In this work, we seek to address the adverse properties of Rsg, such as solubility, by the technique of crystal engineering. A search of literature including the Cambridge Structure Database (CSD) found that the crystal form reports of rosiglitazone was mostly related to its acid salt forms, while the co-crystallization with other drugs was rarely reported. Therefore, it is worthy to investigate drug-drug co-crystals based on Rsg and another APIs. Because of the excellent therapeutic effect on type 2 diabetes and the water solubility of its chloride, we decided to choose the Bbr to be a co-crystal former of rosiglitazone. 

By analyzing the structure of two compounds, we found that the N-H bond on the thiazolidinedione of rosiglitazone easily loses protons to become a nitrogen anion with the influence of two adjacent carbonyl groups. It should be noted that berberine, a quaternary ammonium cation, contains a positive charge without acquiring a proton, unlike amine cations. Considering the compound charge stability, the cationic nature of berberine requires the presence of charge equivalent anion to maintain charge neutrality. Thus, we speculate whether the nitrogen anion on the thiazolidinedione ring can be substituted for the chloride ion in berberine chloride to allow rosiglitazone and berberine to form a novel adduct through non-covalent bonds. In this paper, the co-crystallization of rosiglitazone and berberine was carried out by the solvent volatilization method. As a result, the co-crystal Rsg-Bbr has successfully been prepared. Its structure was characterized by Fourier-transform infrared (FT-IR), powder X-ray powder diffraction (PXRD), thermogravimetry (TGA), differential scanning calorimetry (DSC), single crystal X-ray diffraction (SCXRD), and polarized optical microscopy (POM). The dissolution rate and hygroscopicity of the co-crystal were also explored.

## 2. Results and Discussion

### 2.1. The Characterization Analysis of the Co-Crystal Structure

With the anion exchange reaction and the simple method of solvent evaporation, Rsg-Bbr was successfully prepared in the methanol within five days. The FT-IR and PXRD were used for the preliminary detection of the difference between Rsg-Bbr and two raw materials.

Infrared spectroscopy is a simple and rapid test method in the co-crystal screening experiment, it can preliminarily identify the difference between the co-crystal and two materials by comparing with individual components. In the infrared spectra of the Rsg-Met (Figure 2), the N-H stretching vibration (3379 cm^−1^) of Rsg disappeared, which was influenced by the removal of the H atom of the amino group makes nitrogen become anionic. The thiolactone C=O vibration (1693 cm^−1^) of Rsg moved to low-frequency and displayed a novel absorption peak at 1681 cm^−1^, which was influenced by the formation of N anions, and the intermolecular hydrogen bonds. Furthermore, compared with the corresponding groups in the pure Bbr material, the C=N vibration peak (1634 cm^−1^) moved to low-frequency and showed a novel absorption peak at 1598 cm^−1^, which was affected by the electrostatic interaction. The C=C of benzene rings stretching vibration peak (1600 cm^−1^) moved to low-frequency with the influence of the π-π effect between molecular layers and displayed a novel absorption peak at 1562 cm^−1^. These results indicated that the intermolecular interaction could cause some subtle changes in the molecular structure. 

Powder X-ray diffraction pattern of Rsg-Bbr showed a new characteristic peak by comparing with two monomer components (Figure 3). The characteristic peaks (two theta in degree) of the reactants and product were as following, Rsg (15.2, 15.6, 17.4, 18.2, 20.1, 22.4, 32.2), Bbr (6.5, 9.18, 10.71, 13.14, 25.15, 25.56, 26.71) and Rsg-Bbr (16.3, 20.9, 25.01, 25.5, 30.6, 30.8). Additionally, the experimental PXRD patterns of Rsg-Bbr well matched the simulated PXRD patterns of corresponding single crystal structure. This confirmed the phase purity of the bulk powders. 

The data of this co-crystal measured by single crystal X-ray diffraction (SCXRD) experiment further confirmed the formation of Rsg-Bbr. The corresponding crystallographic data and refinement details were summarized in Table 1. The crystal structure of Rsg-Bbr was in the triclinic space group P−1: a = 7.4411(4) Å, b = 13.3185(6) Å, c = 18.8457(10) Å, α = 98.950(4)°, β = 98.400(4)°, γ = 101.178(4)°, Z = 2, and the final R_1_ was 0.0584 (I > 2σ(I)) and wR_2_ was 0.1833 (all data). The asymmetric unit of the solid is shown in Figure 4, it contained one molecule of Rsg, one molecule of Bbr, and one molecule of methanol.

However, only weak D-H···A hydrogen bonds (Table 2) were present to stabilize this structure. As shown in Table 2 and Figure 5, there existed obvious hydrogen-bonding interactions between Rsg and Bbr. Specifically, Bbr established the hydrogen bonding by the 11-H atom and 14-H-atom on the B-pyridine ring with the lactam group and the thiolactone group of the Rsg molecule, respectively C(11)-H(11)···O(6) and C(14)-H(14)···O(5). 5-carbonyl group of Rsg was involved in intramolecular hydrogen-bonding interactions with the H-D^b atom on 24-methlene group [C(24A^b)-H(24D^b)···O(5)]. As a solvent, methanol interacted with rosiglitazone ion and berberine ion by forming hydrogen bonds O(8A^b)-H(8AA^b)···N(4A^b) and C(7)-H(7)···O(8^a), respectively. In addition, the electrostatic attraction between the nitrogen anion of rosiglitazone and the quaternary ammonium cation of berberine made a contribution to stabilize the crystal structure.

When the crystal structure elements were repeatedly arranged in three-dimensional space to form a co-crystal, a layer structure was generated in the structure, and π electron interacted between layers. Unfolded the unit cell diagram, we found that the π-π effect existed between the molecular layers. In Figure 6, there existed various π-π interactions between benzene rings A and C of berberine. Berberine cation simultaneously interacted with two adjacent berberine through self-complementary π-π interactions between rings A and C. The distance of centroid···centroid was 3.854 Å and 3.759 Å, respectively. A one-dimensional supramolecular layer of berberine resulted from the π-π interactions. Rosiglitazone contained two non-coplanar A-phenyl ring and B-piperidine ring in its molecular structure, and there existed various π-π interactions between rosiglitazone molecules. Ring A of rosiglitazone was involved in π-π interactions with ring A of an adjacent rosiglitazone in the self-complementary way with a distance of 3.6604 Å (centroid···centroid). Two adjacent rings B were also π-π stacked in a self-complementary manner, with a distance of 3.5839Å. These π-π interactions combined with the C(24A^b)-H(24D^b)···O(5) between the rosiglitazone molecules to form a two-dimensional rosiglitazone layer (Figure 7). As a result, with non-covalent bonding forces such as hydrogen bonding, electrostatic attraction, and π-π interaction, the crystal packing structure of Rsg-Bbr was formed.

Thermal analysis is often used to characterize the structure and stability of materials. Thermogravimetric analysis (TGA) and differential scanning calorimetry (DSC) were employed to investigate the thermal properties of Rsg-Bbr, Rsg, and Bbr samples. As shown in Figure 8a, the melting point of this co-crystal was between Rsg and Bbr, which their exothermic peaks accompanied by chemical decomposition at 172, 157, and 193 °C, respectively. Furthermore, the TGA thermogram of Rsg-Bbr (Figure 8b) showed a slight but continuous weight loss of 4.42% before its melting point. This indicated the slow elimination of methanol molecules from the holes formed between the molecular layers. What’s more, the weight of Rsg-Bbr did not change in the temperature range 120 to 210 °C. Consequently, we can conclude that the substance has thermal stability in 120–210 °C.

To observe the changes in the morphology of crystal, a polarizing optical microscope (POM) was used to observe the production of Rsg, Bbr and Rsg-Bbr. Each sample was observed at a magnification of 40×. It was obvious that the shape of Rsg-Bbr crystal was distinguishable from the two raw materials. This test further confirmed the presence of Rsg-Bbr adduct (Figure 9).

### 2.2. The Characterization Analysis of Crystal Dissolution and Hygroscopicity 

Dissolution rate is one of the key pharmaceutical properties that need to be considered for successful drug delivery. Taking into account that the pKa of rosiglitazone is about 7, which is a weakly basic drug and assimilated mainly in the intestine of the human body. Therefore, the phosphate buffer solution with a pH of 6.8 was used as the dissolution medium. According to Figure 10a below, the Rsg-Bbr group reported significantly and consistently higher dissolution than Rsg alone during the whole period. In the end, it peaked at the highest point of around 59%. Due to the poor solubility of Rsg, its dissolution rate finally was less than 10%. Notably, we also compared the dissolution rates of berberine and Rsg-Bbr. The dissolution rate of Bbr was significantly higher than that of the Rsg-Bbr in Figure 10b. Their highest value of rates achieved about 73% and 51% respectively. Compared the rate values of these samples, the trend depicting the dissolution rates of them was as follows: Bbr > Rsg-Bbr > Rsg.

Dynamic vapor sorption (DVS) analysis was performed to compare the hygroscopicity of Rsg-Bbr with Rsg. The sensitivity of a drug to moisture is one of the factors that affect its drug properties. The medicinal efficacy is affected badly by the moisture. According to reports, the berberine chloride has a hygroscopicity close to 20% in a high humidity environment with a relative humidity of 95%, which is a medium hygroscopic substance [24]. As exhibited in Figure 11a, Rsg did not begin to absorb 0.46% water until the relative humidity reached 70%. This result was expected because Rsg has poor solubility and hence low hygroscopicity. In Figure 11b, Rsg-Bbr presented low moisture and 0.56% of water in the range of 0–95% RH at 25 °C. This indicated that the co-crystal was non-hygroscopic. In addition, the desorption curve was close to the absorption curve for Rsg-Bbr, indicating the absorption and desorption processes of the co-crystal was reversible [10].

## 3. Materials and Methods 

### 3.1. Materials

Rosiglitazone was purchased from Adamas Reagent Company (Shanghai, China). Berberine chloride (West plant extraction factory, Sichuan, China) was used as received. Other chemicals were purchased from Adamas Reagent Company and used without any further purification.

### 3.2. Methods of Synthesis

Synthesis of Rosiglitazone Sodium: Sodium metal (0.07 g, 0.003 mol) was shredded and added to 15 mL of anhydrous ethanol to make be sodium ethoxide. Rsg (1.07 g, 0.003 mol) was dissolved in 80 mL of ethanol by heating at 70 °C. Pouring into sodium ethoxide after the solution of Rsg was clarified. Then, the whole reaction system refluxed for 10 h. Finally, filtered the suspension to get the white powdery rosiglitazone sodium.

Synthesis of Rsg-Bbr: Berberine chloride (1.17 g, 0.003 mol) and the prepared sodium rosiglitazone salt were input 60mL of methanol to dissolve. Then, heated to 70 °C and refluxed for 1 h. The obtained yellow filtrate was placed at 4 °C. Crystals suitable for single crystal X-ray diffraction experiment was obtained in 5 days.

### 3.3. Methods of Structural Analysis

#### 3.3.1. Fourier Transformation Infrared Spectroscopy (FT-IR)

FT-IR spectra were collected by a Nicolet Spectrum FT-IR spectrometer (Nicolet iS10, Waltham, MA, USA) in the range from 4000 to 500 cm^−1^, with a resolution of 4 cm^−1^ at ambient conditions. Each compound (1 mg) was ground into the powder with dried KBr (50 mg) in a mortar and the mixture was pressed into a piece of slice. 

#### 3.3.2. Powder X-ray Diffraction (PXRD)

Powders were analyzed on a Bruker D8 Advance X-ray diffractometer (Bruker, Karlsruhe, Germany) with Cu Kα radiation (1.54056 Å), with two-theta pre-calibrated using a silicon standard. Data for each sample were collected from 5 to 50° two theta with a step size of 0.02° at 1 s/ step and at 25 °C with step and scan speed of 5°/min. The tube voltage and amperage were at 40 kV and 100 mA, respectively.

#### 3.3.3. Thermal Gravimetric Analysis (TGA) 

Samples (10 mg) were heated in a hermetically sealed aluminum pan containing a pinhole, on a NETZSCH STA 449 C (NETZSCH, Selb, Germany) from room temperature to 500 °C at 10 °C/min under 50 mL/min dry nitrogen purge.

#### 3.3.4. Differential Scanning Calorimetry (DSC) 

Differential scanning calorimetry can detect the inherent melting point of each crystal. Powder samples (5 mg) were heated from 25 to 250 °C with a heating rate of 10 °C/min on a NETZSCH-TA4 STA Instruments 449C differential scanning calorimeter (NETZSCH, Selb, Germany) under a continuously purged dry nitrogen atmosphere (flow rate of 50 mL/min). Tzero hermetic sealed aluminum pans were used for all samples. The instrument was pre-calibrated for temperature and enthalpy using indium.

#### 3.3.5. Single Crystal X-ray Diffraction (SCXRD)

Single crystal X-ray diffraction was carried out on a SMART CCD diffractometer (Bruker, Karlsruhe, Germany). The data collection was performed at 293 K using CuKα radiation (λ = 1.54184 Å, graphite monochromator). The integrated and scaled data were empirically corrected for absorption effects with spherical harmonics, implemented with the Sadabs scaling algorithm. Using Olex2 [25], the structure was solved by the ShelXS [26] structure solution program using direct methods and refined with the ShelXL [27] refinement package using least squares minimization. All non-hydrogen atoms were refined with anisotropic displacement parameters. All hydrogen atoms were located from the difference Fourier map and allowed to ride on their parent atoms in the refinement cycles.

#### 3.3.6. Polarized Optical Microscopy (POM)

Several droplets of Rsg, Bbr and Rsg-Bbr are deposited on the microscope slide and sliced for stably observed their crystal morphology. These samples were optically characterized at 25 °C by POM using an Olympus transmission microscope coupled with a Leica digital camera and Leica Application Suite Software.

### 3.4. Methods of Physicochemical Properties

#### 3.4.1. Dissolution Rate 

For Rsg, Bbr and the co-crystal, the dissolution rate was performed by using a US Pharmacopoeia tablet dissolution test apparatus (paddle method, Hanson Research, Chatsworth, CA, USA) in 900 mL of phosphate buffer (pH 6.8) containing 0.2% (*w*/*v*) of tween-80 as a dissolution medium. The rotation speed was set at 50 rpm with dissolution bath temperature of 37 °C. Pure Rsg, Bbr and Rsg-Bbr powder were added to dissolution vessel containing 900 mL of phosphate buffer 6.8 pH. Aliquots of the dissolution medium (5 mL) were withdrawn at 10, 20, 30, 45, 60, 80, 100, 120, 160, 180 and 240 min. The dissolution media was replaced with an equivalent volume of fresh media at 37 °C. 4 mL of the secondary filtrate was taken by filtering with 0.45 µm filter (Titan, Shanghai, China) and discarding the primary filtrate. The UV-Visible detector (UV2600, Shimadzu, Kyoto, Japan) were set at two wavelengths of 235 nm and 345 nm (the wavelength of 235 nm to detect Rsg, the wavelength of 345 nm to detect Bbr). Each sample is balanced for 3 times. At last but not least, the data were detected by UV-Visible detection and linear regression were observed in the concentration (C, µg/mL) by peak area (A) that equation were: A_Rsg_ = 0.0267C + 0.0114 (r = 0.9999) and A_Bbr_ = 0.018C + 0.0861 (r = 0.9998). The value of the dissolution rate calculated by the equation w = CV/m × 100% was used to draw, ultimately. 

#### 3.4.2. Dynamic Water Vapor Sorption Isotherm (DVS) 

Water sorption and desorption profiles of the materials were measured by using an automated vapor sorption analyzer (SMS Ltd., London, UK) at 25 °C. Samples of Rsg-Bbr and Rsg were studied at each step with the equilibration criteria of either dm/dt ≤ 0.003% (the defining time is 5 min) or maximum equilibration time of 6h. Once one of the criteria is met, the relative humanity (RH) was changed to the next target value, following the 0%–95%–0% sorption and desorption cycle with the step size of 5% RH.

## 4. Conclusions 

The co-crystal of rosiglitazone (Rsg) with berberine chloride (Bbr), Rsg-Bbr, was created and characterized. As far as we know, this is the first report to show a new drug-drug co-crystal formed by Rsg and a natural drug with a hypoglycemic effect. The preparation process was simple with easy handling. Crystal structure analysis showed that the nitrogen anion of rosiglitazone successfully has replaced the chloride ion in berberine chloride to form the co-crystal. The electrostatic attraction was also generated between the anion and cation. Moreover, this electrostatic interaction combined with the C-H···O hydrogen bond formed between Rsg and Bbr and the π-π interaction formed between conjugated rings to dominate the crystal packing structure. 

Additionally, Rsg-Bbr exhibits a low moisture adsorption curve in the range of 0–95% relative humidity at 25 °C. Rsg in the form of the co-crystal has a higher dissolution rate compared with pure rosiglitazone. The work described above shows a promising method that can be used to overcome the poor physicochemical properties of the parent drugs by forming the co-crystal structure. The pharmaceutical crystals based on drug-drug combinations could contribute to the development of new drugs in the pharmaceutical field.

## Figures and Tables

**Figure 1 molecules-25-04288-f001:**
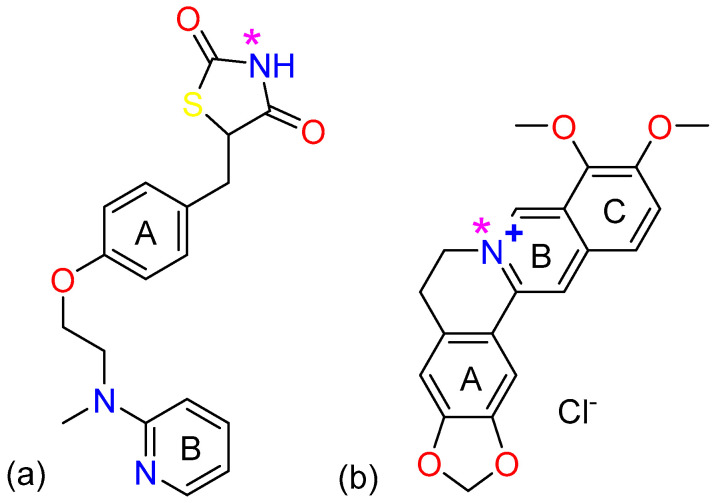
Chemical structures of the Rosiglitazone (**a**) and Berberine chloride (**b**) with the N-H and quaternary ammonium cation sites highlighted, respectively.

**Figure 2 molecules-25-04288-f002:**
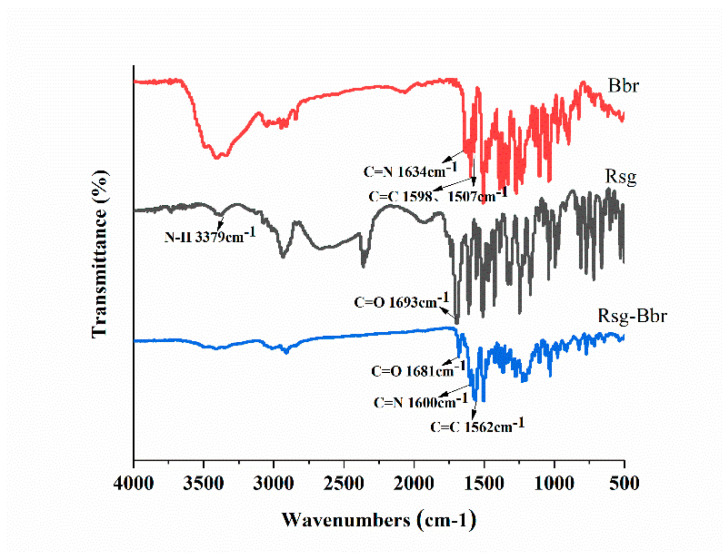
IR spectral data for **Bbr**, **Rsg**, and **Rsg-Bbr.**

**Figure 3 molecules-25-04288-f003:**
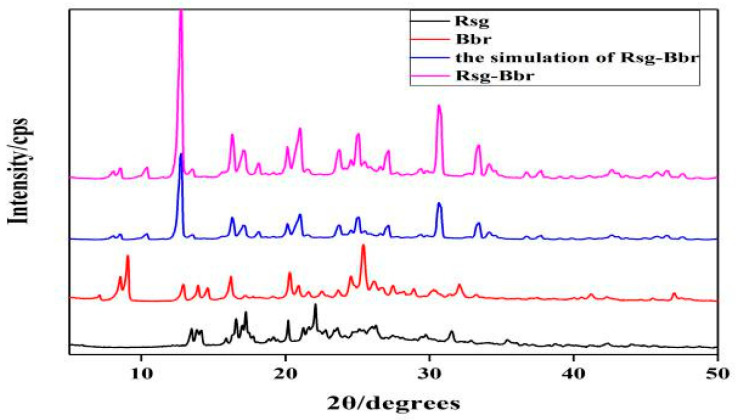
X-ray powder diffractograms of Rsg-Bbr, Rsg and Bbr.

**Figure 4 molecules-25-04288-f004:**
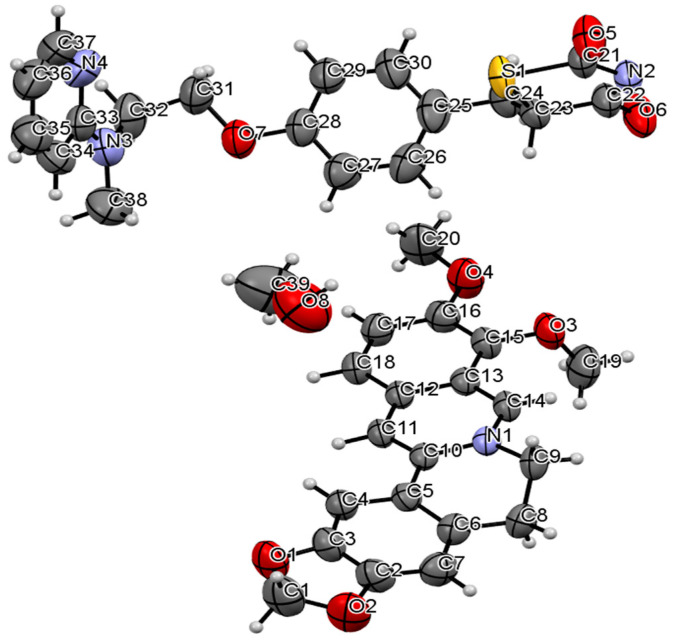
The ORTEP diagram of Rsg-Bbr.

**Figure 5 molecules-25-04288-f005:**
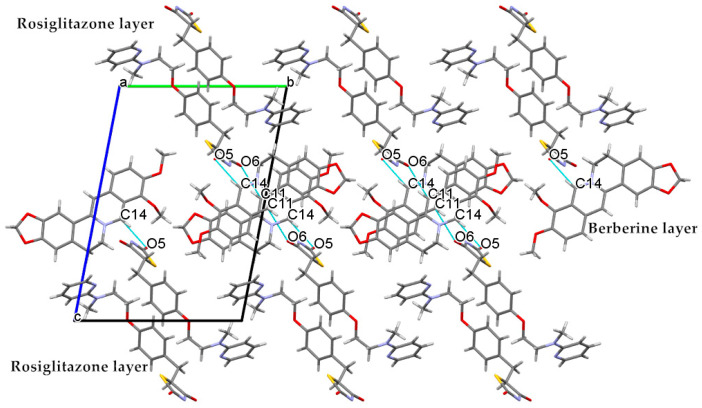
The unit cell diagram of Rsg-Bbr viewed along a-axis. Dashed lines represent hydrogen-bonding interactions between Rsg and Bbr.

**Figure 6 molecules-25-04288-f006:**
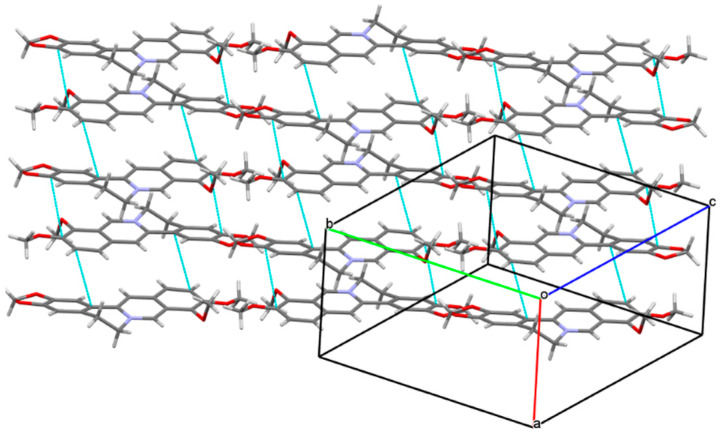
One-dimensional supramolecular layer Bbr resulting from π-π interactions.

**Figure 7 molecules-25-04288-f007:**
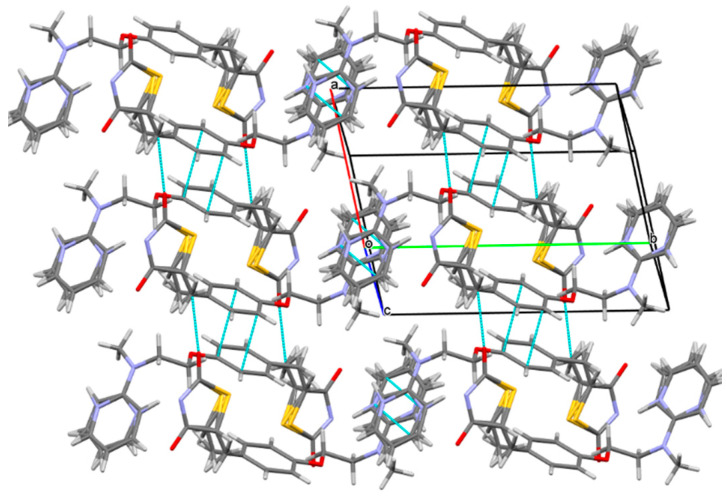
Two-dimensional supramolecular layer Rsg resulting from π-π and C-H···O interactions.

**Figure 8 molecules-25-04288-f008:**
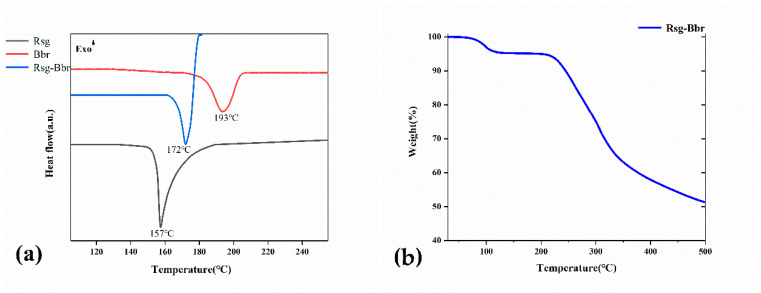
(**a**) DSC thermograms of Rsg, Bbr and Rsg-Bbr, respectively. (**b**) TG thermogram of Rsg-Bbr.

**Figure 9 molecules-25-04288-f009:**
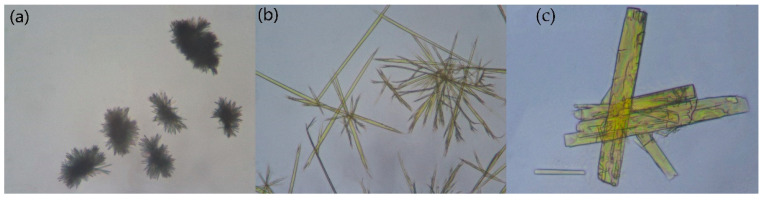
Micrographs of samples obtained by POM at 25 °C. (**a**) The crystal form of Rsg is rod-shaped (**b**) The crystal form of Bbr is needle-shaped (**c**) The crystal form of Rsg-Bbr is prismatic-shaped.

**Figure 10 molecules-25-04288-f010:**
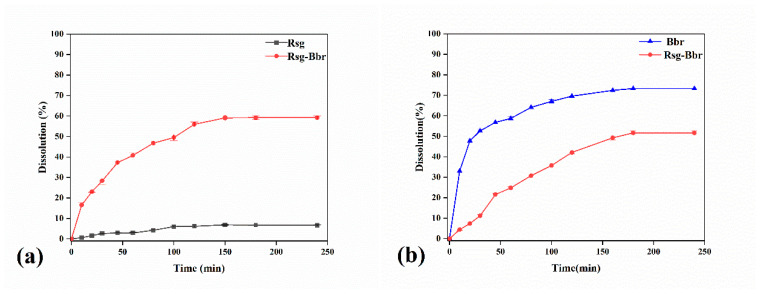
(**a**)The dissolution rate of Rsg and Rsg in the co-crystal; (**b**)The dissolution rate Bbr and Bbr in the co-crystal, respectively (pH = 6.8).

**Figure 11 molecules-25-04288-f011:**
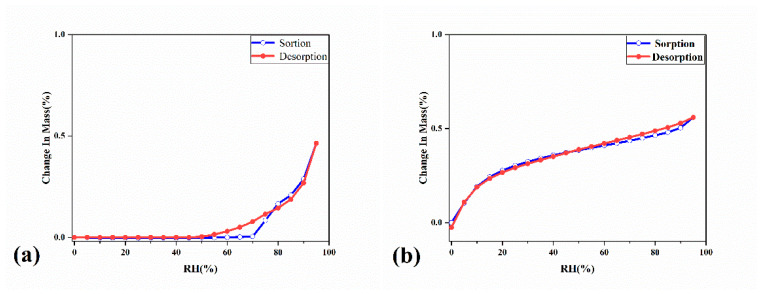
(**a**) DVS isotherm plots for co-crystal of Rsg-Bbr; (**b**) DVS isotherm plots for raw material of Rsg at 25 °C.

**Table 1 molecules-25-04288-t001:** Crystal Data and Structural Refinement for Rsg-Bbr.

Empirical Formula	C_39_H_40_N_4_O_8_S
molecular weight	724.81
Crystal size/mm^3^	0.22 × 0.15 × 0.12
Crystal system	triclinic
Space group	*P* − 1
a/Å	7.4411(4)
b/Å	13.3185(6)
c/Å	18.8457(10)
α/°	98.950(4)
β/°	98.400(4)
γ/°	101.178(4)
V/Å^3^	1779.75(16)
Z	2
ρ_calc_g/cm^3^	1.353
μ/mm^−1^	1.307
F(000)	764.0
2θ range for data collection/°	7.628 to 134.16
Index ranges	−8 ≤ h ≤ 8, −15 ≤ k ≤ 10, −22 ≤ l ≤ 22
Reflections collected	12524
Independent reflections	6335 [R_int_ = 0.0261, R_sigma_ = 0.0361]
Data/restraints/parameters	6335/53/496
Goodness-of-fit on F^2^	1.038
Final R indexes [I ≤ 2σ (I)]	R_1_ = 0.0584, wR_2_ = 0.1635
Final R indexes [all data]	R_1_ = 0.0771, wR_2_ = 0.1833
Largest diff. peak/hole / e Å^−3^	0.40/−0.26
CCDC no.	2007762

**Table 2 molecules-25-04288-t002:** Hydrogen bonding table for the co-crystal Rsg-Bbr.

D-H···A	d(D-H)/Å	d(H···A)/Å	d(D···A)/Å	D-H-A/°
C(11)-H(11)···O(6) ^1^	0.93	2.63	3.453(3)	147.3
C(14)-H(14)···O(5) ^2^	0.93	2.30	3.207(3)	165.8
C(24A^b)-H(24D^b)···O(5) ^3^	0.97	2.45	3.111(16)	125.3
C(7)-H(7)···O(8^a) ^4^	0.93	2.51	3.258(12)	137.9
O(8A^b)-H(8AA^b)···N(4A^b) ^5^	1.05	2.19	3.06(4)	139.6

Symmetry transformations used to generate equivalent atoms: ^1^ x, y − 1, z; ^2^ −x, −y + 1, −z + 1; ^3^ x + 1, y, z; ^4^ −x + 1, −y, −z + 1; ^5^ −x + 1, −y, −z.

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
