# Peer review of "A New Co-Crystal of Synthetic Drug Rosiglitazone with Natural Medicine Berberine: Preparation, Crystal Structures, and Dissolution"

_molecules, 2020, doi:10.3390/molecules25184288_

Round 1
Reviewer 1 Report
This manuscript reported a new co-crystal of Rsg-Bbr to alter the existing adverse properties of single drugs by crystal engineering. Authors also provided a comprehensive characterization analysis to investigate the new Rsg-Bbr co-crystal structures and its physicochemical properties. Such cocrystal engineering technology to improve drug bioavailability and pharmaceutical efficacy is important in drug development. However, in this study, though these two medications (Rsg and Bbr) have been used to treat T2DM, the therapeutic efficacy of the Rsg-Bbr co-crystal remains unpredictable. Would the intermolecular interaction observed in the co-crystals physiologically affect the therapeutic effects? If any evidences from previous studies can support this point, it should be stated more clearly in the introduction section. If not, further pharmacological studies or clinical investigation should be carefully performed. Conclusion section is too short and does not deliver the importance and novelty of their research to the audiences.
Minor Concerns:
Legends of figures 1, 2, 9, and 10 should be stated with more details.
From line 132 to 137, text and typography are very different from other paragraphs.
Figure 8: missing scale bars in all panels.
Figure 9: why is the unit for Time defined as “%”?
Dissolution rate and sensitivity of drug to moisture were investigated in this study. However, no dissolution rate of Bbr was listed as a control in figure 9; sensitivity of Rsg to moisture was not mentioned in the text, no previous results shown neither. If these physicochemical properties were considered as key factors for successful drug delivery, all raw chemicals (Rsg and Bbr) should be investigated in both analyses.
Line 212, in the DSC analysis, heating temperature was from 25 to 250 °C, which is not consistent with the range shown in Figure 7a.
Line 228: “BER” should be change to “BBR”?
Line 238: what does “lowered into” mean?
Line 241: does “Four mL” mean 4 mL?
Author Response
Please the attachment

Reviewer 2 Report
The authors examined the crystal structure analysis of Rsg-Bbr, which is a cocrystal of two drugs of controlling blood sugar (Rsg anion) and anti-inflammatory (Bbr cation), and studied the stability and solubility of the crystals. The crystal structure determination was well carried out. Such research will be of interest to researchers in the fields of organic chemistry, biology, as well as crystallography and pharmacology. The reported field is sufficient for the article of ‘Molecules’. However, I cannot recommend the publication of this manuscript without major revision because the descriptions are not enough for several texts and figures.
My comments are as follows,
1) The authors will have to ask a native English speaker to read the manuscript, and correct the numerous grammatical and spell errors.
2) This is hard to get readers to understand because the research story is not well and enough.
・Figure 3 should be inserted in Introduction (in Page 2). To understand the research for many readers, the molecular structures of Rsg and Bbr are necessary in the first sentence. Please show the charge information and the relationship of the starting material of BbrCl. In page 2, line 45, please give the more information of Bbr cation because the present description of Bbr makes confusing the readers.
・Before discussion of IR, the corresponding preparation comments are requested for understanding the sample quality and solvate information.
3) For three Figures 2, 4, and 5, the crystal structure is not representative of the molecular structure, stoichiometry, and intermolecular interactions. Because the main discussion should focus the structure and reason of the association of two different molecules in the cocrystal, please consider the figures from the beginning.
・Figures 2 and 5 are important for understanding the cocrystal, but both graphics are low quality. Why did the authors use black and white and a collapsed aspect ratio for the important Figure 2 even they use color and better quality of Figure 4?
・In Figure 4, it's important to see the interaction for the arrangement with remarkable instructions, e.g., dash lines, and the axes of unit cell.
・The authors should show the whole packing structure of Figure 5 before Figure 4. The graphics should be improved for understanding the two different molecules. No information was obtained from the present style.
4) Why did the authors use a MeOH solvents for the crystallization? For considering the life-toxicity of MeOH, it is necessary to crystallize it with EtOH or water. Please add the comments for the strategy and comparison of the crystallization solvents.
In addition, please check as follows,
1) Page 1, line 33: sacrabitol? or sacubitril?
2) Page 2, lines 61-63: Please improve the text to make it more understandable.
3) Page 2, lines 67-69: Please improve the text to make it more understandable.
4) Page 2, line 77: For IR, the authors use the BbrCl hydrate for the reference. Please describe correctly.
5) Page 4, line 100: Why ‘surprisingly’? The molecules, Rsg and Bbr, have no site of D-H. Because the hydrogen bond is not between Rsg-Bbr and it’s between MeOH and Rsg, please revise the text and the caption of Table 2.
6) Figures 6, 7a, and 8: Please align the data order and the annotation order in the same direction.
7) Page 6, line 145 and Figure 7b: Please add the weight loss % with both experimental and calculation values. In Figure 7c, please focus the weight loss (%) to 100-40.
8) Page 6, lines 146-147: Is "better thermal stability" a general statement? Please describe the details.
9) Figure8c: ‘column-shaped’ is not common for the description of crystals and ‘prismatic’ is better.
10) Page 7, line 159: Please correct to an appropriate title.
Reviewer 3 Report
This paper is an interesting contribution to co-crystallization, in this case of two different drug molecules. The experimental work seems well carried out, but I have a few comments that I would like the authors to consider. However, first I need to ask the authors to improve the english presentation, it is urgently required, as it in some cases prevents the understanding and generally moves focus from content to style.
My questions that the authors need to reply to before acceptance are:
Line 14: "reliable electrostatic attraction". What does that mean? I dont see the authors anywhere show any quantities that says anything about the degree of attraction between molecules!
Line 25: eutectic is a word that relates to a melt, but in this case there is no melt. In addition, in eutectic melts the two components are higher melting than the mixture, but this is not so here!
Line 47: "known to all" is not correct, I am sure there are many more than me that does not know the following statement in that line.
Figure 2 is of poor resolution. I dont see any ammonium as mentioned??? Please indicate where that is, and can you confirm it is found by X-ray analysis? I need the cif file, please provide that for me.
line 100: What is a classic hydrogen bond?? Do you mean "classical"? Probably, yes, but still what does that mean? I am sure there are more C-H...X hydrogen bonds in the world than O-H...O hydrogen bonds, so why are some classical?
line 103: "cation in the Bbr is the main factor for crystal formation" how can you possibly know this? I dont see any evidence for that statement? It may be correct but you have not shown it, as far as I can see.
Table 2: In the column d(D-H), you give numbers but these are obtained from Shelxl, and have not been measured, is that correct? You need to describe that in the experimental section, or remove it from this table.
Figure 4: I am surprised to see in Figure 4a three different labelling schemes. Are there three molecules in the asymmetric unit?? Or are these symmetry related molecules?
line 133: why do you mention that the cocrystal has different PXRD pattern than the monomers? This is obvious from all the rest.
line 146: What is a crystal hole? I have not heard about this before?
Figure 8. The Rsg crystals are not cluster shaped. The clusters are agglomerates of what seems to me to be rod or needle shaped tiny crystals.
line 221: Why do you use SCALE3 ABSPACK with a Bruker diffractometer, where you would normally use Sadabs?
Round 2
Reviewer 2 Report
I would like to recommend the current manuscript to 'Molecules' as it had undergone major revisions.
Author Response
Dear Professor:
Thanks for your comments and suggestions on our manuscript. Secondly, thank you very much for your recognition of our revision, and I’m also very grateful to you for recommending our manuscript to "Molecules". Finally, we appreciate for your warm work earnestly.
Best wishes!
Reviewer 3 Report
The changes are satisfactory.
Author Response
Dear Professor:
Thanks for your comments and suggestions on our manuscript. I am very grateful to you for your recognition of our revision. I read through the manuscript again and found some language problems. I have tried my best to tackle these problems and sent the revised manuscript to the editor. Finally, we appreciate for your warm work earnestly.
Best wishes!